# A Randomized Comparison between 4, 6 and 8 mL of Local Anesthetic for Ultrasound-Guided Stellate Ganglion Block

**DOI:** 10.3390/jcm8091314

**Published:** 2019-08-27

**Authors:** Yongjae Yoo, Chang-soon Lee, Yong-Chul Kim, Jee Youn Moon, Roderick J. Finlayson

**Affiliations:** 1Department of Anesthesiology and Pain Medicine, Seoul National University Hospital, Seoul National University College of Medicine, Seoul 03080, Korea; 2Integrated Cancer Management Center, Seoul National University Cancer Hospital, Seoul 03080, Korea; 3Department of Anesthesia, Alan Edwards Paint Unit, McGill University Health Center, 1650 Cedar Ave, Montreal, QC H3G 1A4, Canada

**Keywords:** complex regional pain syndrome type I, stellate ganglion block, temperature asymmetry, sympathetically maintained pain

## Abstract

Background: Because it affords greater accuracy than landmark-based techniques, ultrasound guidance may reduce the volume of local anesthetic required for sympathetic blockade of the upper extremity. We hypothesized that 4 mL would provide a similar clinical effect when compared to larger volumes. Methods: One hundred and two patients with chronic neuropathic pain of the upper extremity or face were randomly assigned to receive an ultrasound-guided (USG) stellate ganglion block (SGB) with either 4 mL (group A), 6 mL (group B) or 8 mL (group C) mL of 1.0% lidocaine. Skin temperatures of the face, hand, and axillary fold were measured bilaterally at baseline, 10, 20, and 30 min after the block. Our primary outcome was the relative increase in hand temperature on the blocked side at 30 min and our non-inferiority margin was −0.6 °C. Secondary outcomes included success rate (as defined by a relative temperature increase of ≥1.5 °C), pain relief, degree of ptosis and side-effects. Results: The 95% confidence intervals for the difference of the means exceeded our non-inferiority margin (A versus B: −0.76 to 0.24; A versus C: −0.89 to 0.11) for temperature changes in the hand; however, success rates were similar (44, 45 and 55% for A, B and C respectively, *p* = 0.651). No intergroup differences were found in temperature-related outcomes for the other measurement sites (face, axilla). The incidence of minor side-effects was significantly higher in group C and no block-related complications were noted. Conclusions: We were unable to establish the non-inferiority of a 4 mL volume for sympathetic blockade of the hand. The clinical significance of these findings is unclear as success rates were similar between the different groups. In contrast, the 6- and 8 mL volumes were not associated with greater temperature changes in the face and axilla.

## 1. Introduction

Stellate ganglion blocks (SGB) are commonly used in the diagnosis and management of sympathetically maintained pain of the face and upper extremity [1,2]. Although the relevant mechanisms are unclear, SGB have been used in a variety of painful conditions such as complex regional pain syndrome types I and II, postherpetic neuralgia, postoperative pain, and atypical facial pain [3,4,5,6,7].

The stellate or cervicothoracic ganglion is formed by the fusion of the inferior cervical and first thoracic ganglions and located in the prevertebral fascia of the C7–T1 vertebra [8]. However, in order to reduce the risk of vertebral artery injury and pneumothorax, SGB are commonly performed at the C6 level [9]. When treating upper limb disorders, injection volumes of 10 to 20 mL have been used during landmark-based techniques to ensure caudal spread to the C7–T1 level [10]. Ensuing Horner’s syndrome, nasal congestion, venodilation and temperature increase in the blocked limb indicate successful blockade [11,12]. Both the landmark- and fluoroscopy-guided techniques have been associated with complications including local anesthetic toxicity and hematoma formation [13].

Ultrasound guidance for SGB was first described more than 25 years ago and improved image quality, as well as greater availability over the last decade, have led to an increasing use of this imaging modality [14]. Because it affords greater accuracy than landmark-based techniques and permits visualization of critical soft tissue structures, it has been speculated that ultrasound guidance could be associated with lower volumes of injectate, in addition to greater safety and efficacy. Previous studies examining the spread of contrast material or the development of a Horner’s syndrome have suggested that volumes from 2 to 7 mL could be used [15,16,17].

However, it remains unclear whether these smaller volumes are associated with successful sympathetic blockade as evidenced by an increase in skin temperature. In addition, their effect on the upper extremity has not been examined. We therefore undertook to examine the effect of ultrasound-guided stellate blocks with three volumes of 1% lidocaine (4, 6 and 8 mL) on the skin temperatures of the hand, axilla and face. Other clinically relevant outcomes such as pain relief, incidence of side-effects, and presence of a Horner’s syndrome were also studied.

## 2. Materials and Methods

This prospective, randomized, patient- and observer-blinded study was approved by the institutional review board of Seoul National University Hospital (IRB SNUH 1705-101-856) and registered with ClinicalTrials.gov (NCT03401801). The manuscript adheres to the applicable Enhancing the Quality and Transparency of Health Research guidelines [18]. Written informed consent was obtained from all participants before their enrollment.

Patients undergoing ultrasound-guided stellate ganglion blocks at Seoul National University Hospital between November 2017 and July 2018, aged 18 to 85 years, with a diagnosis of chronic pain in the upper extremity or face and a numeric pain rating scale score of ≥4 were included. The exclusion criteria were as follows: any peripheral vascular disease in the upper extremity, such as atherosclerosis, embolic disease, vasculitis, Takayasu’s disease, or thoracic outlet syndrome; history of thoracic sympathetic or stellate ganglion neurolysis; coagulopathy; systemic or local infection at the needle injection site; major cervical deformation due to previous surgery; postpneumonectomy status on the contralateral side; known allergy to amide-based local anesthetics; pregnancy and cognitive dysfunction with inability to report pain scores.

### 2.1. Stellate Ganglion Block

Using a random number table obtained from www.randomization.com, patients were assigned to receive an ultrasound-guided (USG) SGB with one of 3 volumes of 1% lidocaine: 4 mL (group A), 6 mL (group B), or 8 mL (group C). Subjects were monitored using electrocardiogram, noninvasive blood pressure monitoring and oxygen saturation throughout the study period. Two anesthesiologists (Y.Y. and C.L.) blinded to the group allocation performed all block procedures in a single room with an ambient temperature of 25 °C. Blocks were performed under sterile conditions with an AFFINITI 50G ultrasound device and a 5–12-MHz linear transducer (Philips Ultrasound, Bothell, WA, USA) using a previously described technique [15]. Patients were place in the supine position with their neck extended and slightly rotated to the contralateral side. The lower part of the neck part of the neck was scanned in a transverse (short axis) plane and the relevant landmarks identified (transverse process of C6, longus colli muscle, prevertebral fascia) (Figure 1). A color Doppler scan was also performed and the position of any vascular structure around the planned needle trajectory was noted (carotid artery, internal jugular vein, vertebral artery, inferior thyroid artery). A 26-gauge, 1.5-inch needle (Korea Vaccine Co., Ltd., Gyeonggi-do, Korea) was inserted via a lateral approach toward the superficial fascia of the longus colli using an in-plane technique. After negative aspiration, the designated volume of 1% lidocaine (Daihan Pharm Co., Ltd., Seoul, Korea) was administered under continuous visualization, with a targeted spread along the anterior aspect of the longus colli and beneath the prevertebral fascia.

Blinding of the operator and patients was ensured by wrapping the syringe in opaque tape and allocation was concealed throughout the data collection process by using an encrypted file identification system. All data was collected by a blinded observer not involved in the patient’s care. The following information was recorded before the block: age, height, weight, smoking status, etiology, comorbid diseases (hypertension, diabetes mellitus, and psychiatric conditions), pain duration, and pain intensity scored using an 11-point numerical rating scale. Skin temperature was then measured on both palms, cheeks, and axillary folds at baseline and then at 10 min, 20 min, and 30 min after injection. To ensure consistency between subjects, the following landmarks were used: a point 3 cm below the palmar aspect of the third metacarpophalangeal joint (hand); at the junction between a vertical line originating from the ipsilateral pupil and a horizontal line originating from the ipsilateral nostril (face) and deepest fold in the axilla. Temperature measurements were recorded at a perpendicular angle and fixed distance of 30 cm using a recently calibrated infrared imaging thermometer (FLIR^®^ TG165TM, The World’s Sixth Sense^®^, Wilsonville, OR, USA) [19]. In addition to pain scores, the grade of ptosis at 30 min was evaluated using the margin reflex distance as follows: none (no eyelid dropping), mild (dropping of 1/4th of the eyelid), moderate (dropping of 1/2 the eyelid), and severe (almost complete dropping) [20]. Patients were also assessed at 30 min and 1 week after the block for possible side-effects and complications (hoarseness, motor weakness in the ipsilateral upper extremity, swallowing difficulty, nausea, vomiting, lightheadedness, seizure, dyspnea, loss of consciousness, evidence of hematoma, infection, recurrent laryngeal nerve injury, phrenic nerve paralysis, pneumothorax and foreign body sensation in the throat).

### 2.2. Statistical Analyses

Our primary outcome was the relative increase in temperature on the blocked side at 30 min. The latter was defined by the following formula: [change in ipsilateral hand temperature (postblock-preblock)]—[change in contralateral temperature] [12]. Based on the findings of previous studies [4,6,16,17], we hypothetized that the 4-mL group would be non-inferior to the 6- and 8 mL ones. Secondary outcomes included the temperature change compared to baseline in the ipsilateral hand, face, and axillary fold at each time point (10, 20, and 30 min). In addition, the reduction in pain scores, degree of ptosis, and block success rates were compared between groups. A successful block was defined by a relative temperature increase of ≥1.5 °C [12].

For sample size estimation, a pilot study (n = 10) was conducted using an 8 mL volume and found an ipsilateral temperature change of 2.00 ± 0.75 °C at 30 min. Demonstrating non-inferiority, would require the 4-mL volume to be compared to the 6- and 8-mL volumes; therefore, the significance level in each test was adjusted to 0.0125 according to the Bonferroni method. To obtain an 80% statistical power with an alpha of 0.0125 and a clinically determined noninferiority margin of 30% (0.60 °C), 31 patients were required in each group (G-power version 3.1.9.2, Heinrich-Heine-University, Düsseldorf, Germany) [21]. Considering a 10% dropout rate, we decided to enroll 34 patients per group.

We adopted per-protocol analysis for the primary and secondary endpoints [22]. Safety analysis was based on the intent-to-treat population. Relative temperature changes at 30 min were examined using the 95% confidence intervals for the difference of the means. Linear mixed models were established to compare temperature changes (°C) in the ipsilateral hand, face, and axillary fold at each time point after the procedure between groups, time points, and groups × time points. To quantify the relationship between the temperature increase in the ipsilateral hand at 30 min and clinical and demographic characteristics, multivariable logistic regression including variables that showed statistical significance (*p* < 0.2) in univariable analysis was performed using a backward method. We also evaluated the association between the temperature increase in the face, hand, and axilla at 30 min and the ptosis grade and change in the pain score using Spearman’s correlation analysis.

The Shapiro–Wilk test was used to assess the normality of continuous variables. Patient characteristics that passed the normality test were compared between groups using one-way analysis of variance and Tukey’s HSD was used for post-hoc comparisons. Other continuous and categorical variables were compared using the Kruskal–Wallis and Fisher’s exact tests, respectively. All statistical analyses were performed using SPSS statistics version 23 (IBM, Armonk, NY, USA). Data are presented as means ± SDs or absolute numbers (%). *p*-values of < 0.05 were considered statistically significant.

## 3. Results

One-hundred-and-nine patients were assessed for eligibility and 102 of them were included in the study (34 per group) (Figure 2). Subsequently, two patients were excluded because of refusal to undergo temperature measurements after the procedure (n = 1 in group B) or the use of a different anesthetic solution (n = 1 in group C). Thus, 100 patients were included in the per-protocol analysis [22]. Demographic variables and patient characteristics are summarized in Table 1.

Relative temperature increases and success rates are presented in Table 2. The lower bound of the 95% confidence interval for the difference of the means exceeded our non-inferiority margin (0.6 °C) for all intergroup comparisons in the hand. In contrast, the margin was not exceeded when comparing group A to groups B and C for measurements in the face and axilla. In addition, no differences were found in success rate at 30 min for any site. Absolute temperatures values by time period and site are presented in Figure 3. Although a significant increase in temperature was noted over time in each of the groups (*p* < 0.001 for all groups and measurement sites), no intergroup differences were found for any of the measurement periods or sites.

Univariable analysis found that body mass index, smoking, diabetes mellitus, psychiatric comorbidities, and patient demographics independently influenced the temperature increase in the ipsilateral upper extremity after ultrasound-guided stellate ganglion block (Table 3). In subsequent multivariable analysis, only body mass index was found to be a significant factor (β = −0.100; 95% CI: −0.177 to −0.022; *p* = 0.013) (Table 3). The incidence and degree of ptosis was similar across groups (Table 4) and no correlation was found between the extent of ptosis and temperature change at 30 min in the ipsilateral hand (Spearman’s correlation coefficient rs = −0.054, *p* = 0.591), face (rs = −0.034, *p* = 0.738), or axilla (rs = −0.058, *p* = 0.567). Post-block pain scores decreased in all three groups, with no intergroup differences (*p* = 0.371) (Table 4). In addition, no association was found between the final temperature increase and the decrease in pain score (linear regression model; *p* = 0.114, 0.294, and 0.159, respectively). Although the incidences of hoarseness and dysphagia were similar across groups, other minor side-effects (transient headache, somnolence or xerostomia) were observed more frequently in group C (*p* = 0.034). No patient displayed sensory or motor changes after the blocks and no major complications were noted.

## 4. Discussion

In this prospective, randomized, observer-blinded trial, we examined the effect of an USG SGB performed with three different volumes of 1% lidocaine. Our study hypothesis was that the 4-mL volume would perform as well as the larger ones. However, we found that the lower end of the 95% confidence interval for the difference of the means exceeded our non-inferiority margin (0.60 °C) in comparisons of relative hand temperature. The size of our non-inferiority margin deserves some discussion. In the absence of any clinical data in the literature to guide our selection, we used hand temperature values derived from our pilot study (2.00 ± 0.75 °C) and reasoned that a variation of 30% (0.60 °C, effect size 0.8) would be meaningful. Although we were unable to demonstrate non-inferiority, no difference in success rates was found between groups. Our definition of success was based on a previous study that had examined the relationship between temperature change and sympathetic block in the hand post SGB [12]. The authors found that 72% of patients presenting a relative temperature increase of ≥1.5 °C had evidence of complete sympathectomy as determined by a negative cobalt blue sweat test [23]. While the relationship between temperature change and sympathetic block has not been explored for other measurement sites (face, axilla), we chose to maintain similar outcome criteria throughout our study and noted comparatively lower success rates at the axilla (23–33%). This may be explained by the fact that the sympathetic supply for this area originates from a lower thoracic segment (T3–T4) [24] and would therefore be less likely to be affected by an injection performed at the C6 level. In contrast, success rates for the hand (44–55%) compare favorably with a previous study that examined the effect of a landmark-based SGB technique using 12 mL of 1% lidocaine and found that only 37% of the patients displayed a relative temperature increase of ≥1.5 °C [25]. Nevertheless, our results echo those of previous studies and indicate that SGBs do not reliably provide complete sympathetic blockade of the hand. While we found no association between the degree of temperature change and extent of pain relief, alternative techniques less likely to be affected by the presence of Kuntz pathways (brachial plexus or thoracic paravertebral blocks) could be considered if the clinical results from a SGB are unsatisfactory [26].

In addition to changes in temperature, the presence of ptosis is often sought as confirmation of block success and we recorded both its presence and extent. Our results indicate that this measure is of limited value in predicting blocks success, as almost all patients (94.1–100%) exhibited ptosis and no association was found between the marginal reflex distance and temperature increase in the ipsilateral hand.

Although our protocol focused primarily on technical factors, patient characteristics can also affect block outcome. Indeed, previous studies have found that diabetes mellitus and smoking can increase peripheral vascular resistance, which could mitigate skin temperature increases associated with sympathetic blockade [27,28]. However, even though both factors showed an association with temperature change in our univariable analysis, they were not significant in a subsequent multivariable analysis. In contrast, we found a significant negative association between BMI and temperature change. Although it is unclear what underlying mechanism might explain this relationship, it has previously been noted for other blocks [29]. Therefore, further studies are needed to investigate the association between those factors and the volume of LA for successful SGB, along with the efficacy of the procedure.

While serious complications related to SGB have been reported in the literature, most events have been transient in nature [13]. Since the majority of these are caused by inadvertent trauma to the various soft tissue structures of the neck and in particular blood vessels, one could speculate that USG would be associated with a lower incidence of adverse events. Our results support this theory, as no major procedure-related complications occurred during the study. Nevertheless, transient side-effects such as hoarseness (11.8–15.2%) and dysphagia (2.9–6.1%) did occur and their incidences were similar to those reported in other studies examining USG SGBs [17]. In addition, other minor events (transient headache, somnolence or xerostomia) occurred significantly more frequently in the 8-mL group. These findings suggest that injections below the prevertebral fascia mays not always prevent aberrant spread of the local anesthetic to unwanted structures and that lower injectate volumes may provide additional benefit. Although we also failed to show the inferiority of 6 mL to 8 mL within our study design, future non-inferiority studies with larger sample size could be considered in terms of the small mean temperature difference observed in this study.

Our study presents some potential limitations. Firstly, while both operators used the same technique, we cannot exclude the possibility that small interindividual variations during block performance could have affected outcomes. Secondly, although the accuracy of infrared thermography for skin temperature measurement has previously been validated [30], its reliability can be influenced by several technical factors [31]. While were able to address the most significant ones in our protocol (measurement site, camera angle and distance), others such as the use of a standardized background, were not possible to implement in our clinical setting and may have had an effect on recorded temperatures. This suggests that other indicators like a sweat test [23] or blood flow [32] may be additively required to confirm complete sympathetic block to the upper extremity by the SGB. Finally, because we recruited patients with a diverse range of painful conditions, the value of our pre- and post-block pain scores as a measure of block efficacy is uncertain.

In conclusion, we were unable to establish the non-inferiority of a 4 mL volume for sympathetic blockade of the hand. The clinical significance of these findings is unclear as the success rate were similar between the different groups. In contrast, the 6- and 8 mL volumes were not associated with greater temperature changes in the face and axilla. The 4 mL volume of LA might not be sufficient for the USG SGB to manage patients with pain in the upper extremity. Although a higher incidence of minor side-effects was observed in the 8 mL group, no complications were noted during the study.

## Figures and Tables

**Figure 1 jcm-08-01314-f001:**
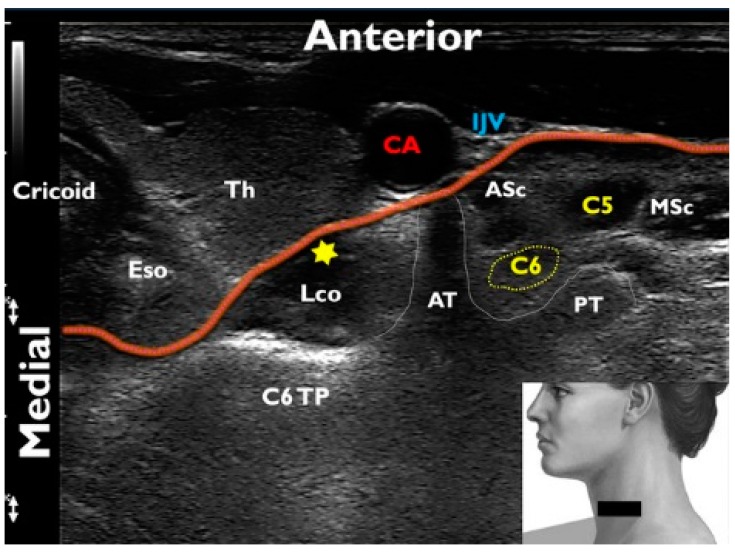
Transverse sonographic view of the neck at the level of C6 for a left stellate ganglion block: The probe placement on the patient’s neck is depicted in the right lower inset; the red line traces the contour of the prevertebral fascia; the yellow star represents the injection target; cricoid cartilage (Cricoid); esophagus (Eso); thyroid (Th); longus colli muscle (Lco); C6 transverse process (C6TP); anterior and posterior tubercles of the transverse process (AT and PT); carotid artery (CA); internal jugular vein (IJV); C6 and C5 nerve roots (C6 and C5); anterior and middle scalene muscles (ASc and MSc).

**Figure 2 jcm-08-01314-f002:**
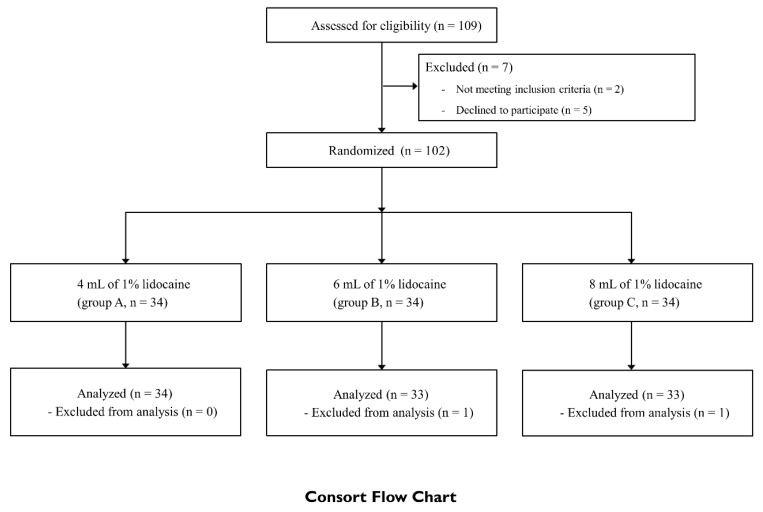
Consolidated Standards of Reporting Trials (CONSORT) flow diagram.

**Figure 3 jcm-08-01314-f003:**
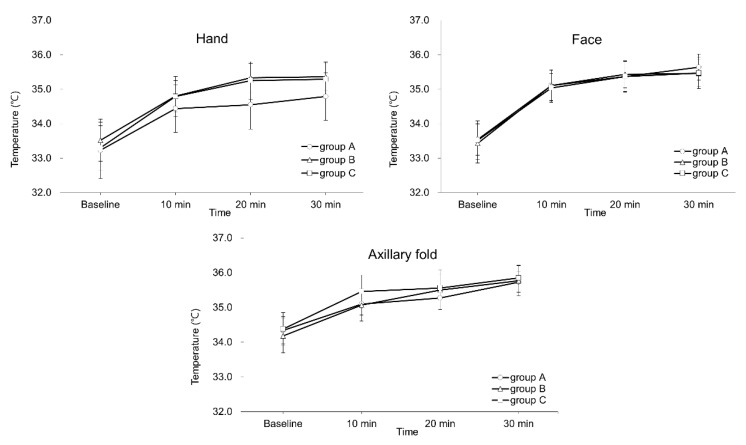
Ipsilateral temperature changes after an ultrasound-guided stellate ganglion block: Charts depict the mean skin temperature changes recorded by time period, for the face, hand and axilla. Groups A, B and C received 4, 6 and 8 mL of 1% lidocaine, respectively. Error bars represent the standard deviation for each data point.

**Table 1 jcm-08-01314-t001:** Baseline Demographic and Clinical Characteristics of Study Participants.

	Total (n = 100)	Group A (n = 34)	Group B (n = 33)	Group C (n = 33)
Sex, n (%)				
Female	68 (68.0%)	22 (64.7%)	21 (63.6%)	25 (75.8%)
Male	32 (32.0%)	12 (35.3%)	12 (36.4%)	8 (24.2%)
Age, years	51.3 ± 13.8	50.7 ± 16.1	50.2 ± 12.8	52.9 ± 12.4
Body mass index, kg/m^2^	24.1 ± 3.9	24.1 ± 3.6	24.6 ± 4.5	23.7 ± 3.6
Smoking, n (%)	11 (11.0%)	5 (14.7%)	2 (6.1%)	4 (12.1%)
Diabetes Mellitus, n (%)	13 (13.0%)	3 (8.8%)	6 (18.2%)	4 (12.1%)
Diagnosis, n (%)				
Face	40 (40.0%)	15 (44.1%)	14 (42.4%)	11 (33.3%)
TN		6	7	2
AFP		2	4	3
PHN		2	1	1
Upper limb	60 (60.0%)	19 (55.9%)	19 (57.6%)	22 (66.7%)
CRPS		5	8	4
PHN		0	0	2
PTPS		3	3	4
Other peripheral neuropathy *		1	0	3
Duration of pain, months	57.1 ± 55.1	51.7 ± 48.1	67.0 ± 48.2	52.3 ± 67.2
Psychiatric comorbidity, n (%)	40 (40.0%)	11 (32.4%)	14 (42.4%)	15 (45.5%)
Laterality, n (%)				
Right	48 (48.0%)	15 (44.1%)	19 (57.6%)	14 (42.4%)
Left	52 (52.0%)	19 (55.9%)	14 (42.4%)	19 (57.6%)
Pre-NRS pain score	6.0 [3.3–8.0]	5.0 [4.0–7.3]	5.0 [3.0–8.0]	7.0 [5.0–8.0]

Data are expressed as mean ± standard deviation, or as median [interquartile range], or as absolute numbers (%). Atypical facial pain (AFN); complex regional pain syndrome (CRPS); postherpetic neuralgia (PHN); PTPS posttraumatic pain syndrome (PTPS); trigeminal neuralgia (TN); numerical rating scale (NRS). * Other peripheral neuropathy in groups A and C include diabetic neuropathy and chemotherapy-induced neuropathy.

**Table 2 jcm-08-01314-t002:** Relative temperature increase and success rate at 30 min by measurement site.

	Mean Relative * Temperature Increase (°C)	Success Rate Ψ N (%)	95% Confidence Intervals for the Difference of the Means
Hand			
Group A	1.24 ± 0.84	15/19 (44.1%)	A versus B: −0.76 to 0.24
Group B	1.50 ± 0.68	15/18 (45.5%)	B versus C: −0.63 to 0.38
Group C	1.62 ± 1.02	18/15 (54.5%)	A versus C: −0.89 to 0.11
Face			
Group A	1.77 ± 0.97	19/15 (55.8%)	A versus B: −0.14 to 1.04
Group B	1.25 ± 0.83	13/20 (39.3%)	B versus C: −0.89 to 0.17
Group C	1.61 ± 0.91	16/17 (48.5%)	A versus C: −0.37 to 0.68
Axillary fold			
Group A	1.00 ± 0.73	8/26 (23.5%)	A versus B: −0.52 to 0.42
Group B	1.04 ± 0.80	12/21 (36.3%)	B versus C: −0.54 to 0.40
Group C	1.12 ± 0.89	11/22 (33.3%)	A versus C: −0.59 to 0.36

(*) relative temperature increase was defined by the following formula: [change in ipsilateral hand temperature (postblock-preblock)]—[change in contralateral temperature]. (Ψ) Success was defined as a relative temperature increase of ≥1.5 °C. All *p* values greater than 0.05 for intergroup comparisons of relative temperature increase and success rates.

**Table 3 jcm-08-01314-t003:** Multivariable analysis between temperature increase and clinical variables (r2 = 0.052)**.**

	Univariable Analysis		Multivariable Analysis	
	B (95% CI)	*p*-Value	B (95% CI)	*p*-Value
Male Age	−0.25 (−0.91, 0.42) −0.01 (−0.03, 0.01)	0.461 0.443	0.02 (−0.77, 0.81) −0.01 (−0.03, 0.01)	0.958 0.288
Body mass index, kg/m^2^	−0.10 (−0.18, −0.02)	0.013	−0.10 (−0.18, −0.02)	0.013
Smoking DM Duration of pain	−0.81 (−1.79, 0.16) −0.72 (−1.63, 0.19) 0.001 (−0.004, 0.01)	0.102 0.119 0.667	−0.73 (−0.17, −0.02) −0.40 (−1.37, 0.57)	0.135 0.417
Location of pain				
Upper extremity	−0.07 (−0.70, 0.57)	0.839		
Laterality (Right)	−0.21 (−0.83, 0.41)	0.505		
Psychiatric comorbidity	−0.46 (−1.09, 0.17)	0.148	−0.32 (−0.98, 0.33)	0.332
Pre-NRS score Group	−0.03 (−0.15, 0.09) 0.22 (−0.16, 0.60)	0.631 0.253		

Backward selection was conducted to retain significant variables (*p*-value < 0.05). B, non-standardized coefficient. Diabetes mellitus (DM); numerical rating scale (NRS).

**Table 4 jcm-08-01314-t004:** Other block-related outcome data.

Group	Group A (n = 34)	Group B (n = 33)	Group C (n = 33)	*p*-Value
Ptosis	32 (94.1%)	33 (100.0%)	33 (100.0%)	0.220
Marginal reflex distance				0.445
None	2 (5.9%)	0 (0.0%)	0 (0.0%)	
Mild	20 (58.8%)	20 (60.6%)	21 (63.6%)	
Moderate	12 (35.3%)	9 (27.3%)	10 (30.3%)	
Severe	0 (0.0%)	4 (12.1%)	2 (6.1%)	
Hoarseness Dysphagia NRS reduction (%)	4 (11.8%) 1 (2.9%) 6.5 [0.0‒30.0]	5 (15.2%) 1 (3.0%) 20.0 [0.0‒30.5]	5 (15.2%) 2 (6.1%) 22.0 [0.0‒50.0]	0.878 0.844 0.371
Other adverse effects *	0 (0.0%)	0 (0.0%)	3 (9.1%)	0.034

Data are expressed as median [interquartile range; P25−P75], or as absolute numbers (%). (*) Transient headache, somnolence or xerostomia. Numerical rating scale (NRS).

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
