# Peer review of "A Randomized Comparison between 4, 6 and 8 mL of Local Anesthetic for Ultrasound-Guided Stellate Ganglion Block"

_jcm, 2019, doi:10.3390/jcm8091314_

Round 1

Reviewer 1 Report

what's the rationale to include a 6 ml group ? why not just 4 vs 8 ml

The result analysis does not clearly states if 6 ml has any value.

What's the final recommendation? keep using higher volume ?

What's the reason there is no correlation between temp change in the face and ptosis ?

Any comment on why despite a good spread on US , the temperature did not increase in many patents in each group , indicating failure to achieve a sympathetic block of the ganglion .

Author Response

REVIEWER #1

First of all, we greatly appreciate your time and effort for your review.

1) Comment #1: What's the rationale to include a 6 ml group? Why not just 4 vs 8 ml?

Response to comment #1:

Previous studies examining the spread of contrast material or the development of a Horner’s syndrome have suggested that volumes from 2 to 7 mL could be used (Jung et al., 2011, Lee at al., 2012, and Yeo at al., 2015). We aimed to show the non-inferiority of lower anesthetic volume and to find the optimal volume for the upper extremity pain. We thought that it was more reliable to compare the 3 groups rather than 2 groups. And we performed USG SGB with the 6ml volume of local anesthetics for the upper extremity pain in our routine process. Thank you very much. 

Changes in manuscript based on comment #1:

No changes made in the manuscript based on comment #1.

2) Comment #2: The result analysis does not clearly states if 6 ml has any value.

Response to comment #2:

In our study, USG SGB was associated with only a small number of mild adverse events. In particular, these symptoms occurred only in patients using 8 mL of LA and no complications were observed in 4 mL and 6 mL Groups. And we found that the mean temperature difference at 30 minutes in hand between Groups B (6 mL) and C (8 mL) was smaller than that between groups A (4 mL) and C(8 mL). Although this result also failed to show the inferiority of 6 mL to 8 mL within our study design, future non-inferiority studies with larger sample size could be considered in terms of the small mean temperature difference observed in this study.

Changes in manuscript based on comment #2:

(Discussion, lines 297-301) These findings suggest that injections below the prevertebral fascia mays not always prevent the aberrant spread of the local anesthetic to unwanted structures and that lower injectate volumes may provide additional benefit. Although we also failed to show the inferiority of 6 ml to 8 ml within our study design, future non-inferiority studies with larger sample size could be considered in terms of the small mean temperature difference observed in this study.

3) Comment #3: What's the final recommendation? keep using higher volume?

Response to comment #3:

We did not show the non-inferiority of temperature increase with 4 ml volume administered in USG SGB. Therefore, the larger volume of local anesthetics might be required for the USG SGB to manage patients with pain in the upper extremity. But considering our limitations, further research is needed.

Changes in manuscript based on comment #3:

(Discussion, lines 313-318) In conclusion, we were unable to establish the non-inferiority of a 4 mL volume for the sympathetic blockade of the hand. The clinical significance of these findings is unclear as the success rate were similar between the different groups. In contrast, the 6- and 8 mL volumes were not associated with greater temperature changes in the face and axilla. The 4 mL volume of LA might not be sufficient for the USG SGB to manage patients with pain in the upper extremity. Although a higher incidence of minor side-effects was observed in the 8 mL group, no complications were noted during the study.

4) Comment #4: What's the reason there is no correlation between temp change in the face and ptosis?

Response to comment #4:

The difference between the direct effect of blocking the sympathetic nervous system and the accompanying effect via vascular dilation may explain the discrepancy. The occurrence of the ptosis after SGB was easily observable but the temperature change in the face was able to be affected by some variables. So, our results indicate that the occurrence of ptosis is of limited value in predicting blocks success and we already mentioned it in the discussion.

Changes in manuscript based on comment #4:

No changes made in the manuscript based on comment #4

5) Comment #5: Any comment on why despite a good spread on the US, the temperature did not increase in many patients in each group, indicating failure to achieve a sympathetic block of the ganglion.

Response to comment #5:

This may be explained by the fact that the sympathetic supply for stellate ganglion area originates from a lower thoracic segment (T3-T4) and would, therefore, be less likely to be affected by an injection performed at the C6 level. And the presence of Kuntz pathways (brachial plexus or thoracic paravertebral blocks) could be considered if the clinical results from an SGB are unsatisfactory. We already mentioned these in the discussion.

Changes in manuscript based on comment #5:

No changes made in the manuscript based on comment #5

Reviewer 2 Report

Dear Authors,

Thank you for your submission. After careful review of your manuscript, here are my comments. The aim and methodology of the study are well defined. Authors have also described their procedure technique adequately. Limitations of the study have also been discussed. Even though the study is interesting, Authors must expand on the conclusion and preferably make it a separate heading. Currently, as a reader, I am confused about the results of the study and is left open to interpretation. Please make a clear recommendation about the results of your study about whether low volume injectate is inferior or not.

-Also, add to your discussion about what needs to be done in the future to improve on the current study to make the results more conclusive.

-Overall a well written manuscript.

Author Response

REVIEWER #2

We very much appreciate your time for reviewing our manuscript. 

1) Comment #1: Thank you for your submission. After careful review of your manuscript, here are my comments. The aim and methodology of the study are well defined. Authors have also described their procedure technique adequately. Limitations of the study have also been discussed. Even though the study is interesting, Authors must expand on the conclusion and preferably make it a separate heading. Currently, as a reader, I am confused about the results of the study and is left open to interpretation. Please make a clear recommendation about the results of your study about whether low volume injectate is inferior or not.

Response to comment #1

We did not show the non-inferiority of temperature increase with 4 ml volume administered in US-guided SGB. Therefore, the larger volume of local anesthetics might be required for the US-guided SGB to manage patients with pain in the upper extremity. But considering our limitations, further research is needed.

Changes in manuscript based on comment #1:

(Discussion, lines 310-315) In conclusion, we were unable to establish the non-inferiority of a 4 mL volume for sympathetic blockade of the hand. The clinical significance of these findings is unclear as success rate were similar between the different groups. In contrast, the 6- and 8 mL volumes were not associated with greater temperature changes in the face and axilla. The 4 mL volume of LA might not be sufficient for the USG SGB to manage patients with pain in the upper extremity. Although a higher incidence of minor side-effects was observed in the 8 mL group, no complications were noted during study.

2) Comment #2: Also, add to your discussion about what needs to be done in the future to improve on the current study to make the results more conclusive.

Response to comment #2

We found a significant negative association between BMI and temperature change. It is unclear what underlying mechanism might explain this relationship even though it has previously been noted for other blocks. Therefore, further studies are needed to investigate the association.

And considering the limitations of infrared imaging thermometer, other indicators like a sweat test or blood flow may be additively required to confirm a complete sympathetic block.

Changes in manuscript based on comment #2:

(Discussion, lines 286-288)

Therefore, further studies are needed to investigate the association between those factors and the volume of LA for the successful SGB along with the efficacy of the procedure.

(Discussion, lines 309-311)

This suggests that other indicators like a sweat test[23] or blood flow[32] may be additively required to confirm complete sympathetic block to the upper extremity by the SGB.